# Comparison of Greater Occipital Nerve Blockade and Sphenopalatine Ganglion Blockade in Patients with Episodic Migraine

**DOI:** 10.3390/jcm13113027

**Published:** 2024-05-21

**Authors:** Hanzade Aybuke Unal, Ahmet Basarı, Opal Sezgi Celiker, Keziban Sanem Cakar Turhan, Ibrahim Asik, Gungor Enver Ozgencil

**Affiliations:** 1Department of Anesthesiology and Reanimation, Division of Pain Medicine, Ankara University School of Medicine, Ankara 06230, Turkey; hanzadeunal@windowslive.com (H.A.U.); dr.ahmetbasari07@hotmail.com (A.B.); iasik@yahoo.com (I.A.); geno6700@hotmail.com (G.E.O.); 2Department of Anesthesiology and Reanimation, Ankara University School of Medicine, Ankara 06230, Turkey; opalsezgi@hotmail.com

**Keywords:** migraine, nerve block, sphenopalatine ganglion block, headache

## Abstract

**Objectives**: Compare the effects of greater occipital nerve (GON) and sphenopalatine ganglion (SPG) blocks on headache intensity and duration, number of headache days, and disability in patients with episodic migraine. **Methods**: In this prospective single-blind randomized study, patients with episodic migraine were randomly divided into two groups: GON and SPG block groups. Patients received blocks once a week for 4 weeks, and once a month for 2 months. The number of headache days, the headache duration, numeric rating scale (NRS) scores, and number of acute medical treatments were assessed before the procedures and 1 month, 2 months, and 3 months after the procedures. Disability was evaluated using the migraine disability assessment (MIDAS) questionnaire at baseline and 3 months after treatment. This study protocol is registered at ClinicalTrials.gov (NCT06243874.). **Results:** 19 patients in the GON block group and 18 patients in the SPG block group were evaluated. Significant improvements in pain severity, headache duration, number of headache days, and the need for acute medical treatment were observed in the 1st, 2nd, and 3rd months compared to baseline in the two groups (*p* < 0.001). There were significant improvements in the MIDAS scores in the third month (*p* < 0.001). The GON block group showed a greater reduction in headache intensity, duration, number of headache days, and MIDAS scores compared to the SPG block group in the 3rd month (*p* < 0.001). **Conclusions**: GON block reduces headache duration, intensity, the number of headache days, and the need for acute medical treatment much more than SPG block in patients with episodic migraine.

## 1. Introduction

Migraine is a neurovascular disease characterized by severe headache, autonomic nervous system dysfunction, and aura in some patients [1]. According to the International Classification of Headache Disorders, third edition (ICHD-3) criteria, a diagnosis of chronic migraine is made when a patient experiences headache on 15 or more days in a month, at least eight of which have the characteristics of migraine. Situations that do not meet the criteria for chronic migraine are referred to as episodic migraine (EM) and are not explicitly defined in the ICHD-3 [2]. In cases with episodic migraine, increases in pain intensity and the number of migraine days indicate a transformation to chronic migraine [3], and so episodic migraine attacks should be treated promptly. Prophylactic treatments for migraine mainly include antidepressants, antiepileptics, beta-blockers, and calcium channel blockers [4,5,6]. However daily medication use may be hindered in cases with cardiovascular/cerebrovascular comorbid diseases, or by drug–drug interactions and renal/hepatic impairment [7]. Due to these conditions, only 23.8% of patients with episodic migraine undergo prophylactic medical treatment [8]. Peripheral blocks as relatively inexpensive, easily applicable, and safe interventions can be applied in cases that fail to respond to medical treatment or that develop side effects. Several studies to date have described the application of blocks to the greater occipital nerve (GON) or sphenopalatine ganglion (SPG) for prophylactic treatment of migraine [9,10,11,12,13,14,15,16,17,18,19,20,21].

The GON originates from the second cervical nerve fibers. There is a convergence of the trigeminal and upper cervical sensory neurons in the trigeminal nucleus caudalis [22,23,24]. The activation of the afferent fibers of the trigeminal nerve stimulates the second-order dorsal horn cells in the trigeminocervical complex and modulates the pain pathways that project the fibers to the thalamus and spinal cord. The antidromic projection of the trigeminovascular system leads to the secretion of the inflammatory vasodilator mediators that are responsible for migraine [25]. It is thought that GON block inhibits the trigeminocervical complex and prevents neurogenic inflammation [9,18].

The SPG, located in the pterygopalatine fossa, contains sensory, parasympathetic, and sympathetic fibers originating in the maxillary branch of the trigeminal nerve, the facial nerve, and the internal carotid plexus and deep petrosal nerve, respectively [26]. The goal of SPG block is to inhibit parasympathetic activation [10], contributing to the inhibition of the perivascular pain receptors in cranial and meningeal blood vessels; and neuroinflammatory mediator release from the sensory fibers that innervate the cranial and meningeal nerves [27]. While there have been several studies evaluating the efficacy of GON and SPG blocks in the treatment of migraine, there has been no study to date comparing their effectiveness. The present study compares the effects of repetitive GON and SPG blocks on headache intensity, duration, number of headache days, and disability in patients with episodic migraine.

## 2. Materials and Methods

Included in this prospective single-blind randomized study were sequentially selected consecutive patients suffering from episodic migraine who presented to the Ankara University Pain Medicine outpatient clinic between September 2023 and December 2023. Episodic migraine was diagnosed in cases where the diagnosis of chronic migraine was excluded according to the International Classification of Headache Disorders, third edition (ICHD-3) criteria published in 2018. Patients experiencing headache days fewer than 15 times a month were classified as having episodic migraine [7]. All of the participants were informed about the study and provided written informed consent for their participation. The study was approved by the local ethics committee (2023/381) and registered on ClinicalTrials.gov (NCT06243874). The study was conducted in accordance with the principles of the Declaration of Helsinki.

### 2.1. Inclusion and Exclusion Criteria

Included in the study were patients aged 18–65 years with a diagnosis of episodic migraine who failed to achieve pain palliation with at least one migraine prophylactic treatment. Due to the preference for medical agents in migraine first-line treatment, all patients had been receiving prophylactic medical treatment for at least six months, and the study consisted of patients who did not show significant changes in headache severity with prophylactic medical treatments. Patients with unstable/severe psychiatric illnesses, pregnancy, bleeding diathesis, known allergies to ingredients given during the procedure (local anesthetics), a history of craniotomy or open skull defects, previous nasal/sinus surgery, medication overuse headache, conditions such as hypertension, vasculitis, malignancy, etc., that may cause headaches, and history of interventional headache treatment such as occipital nerve block, supraorbital nerve block, SPG block, or botulinum toxin injection within the past 6 months were excluded. 

### 2.2. Sample Size and Randomization

The sample size was calculated using a matched pairs *t*-test (two-tailed) with an effect size of 0.5, alpha = 0.05, and power = 0.8, resulting in a total sample size of 34. To control for the potential of missing data, selection bias, and the possibility of drop-outs from the study, it was decided to include 40 people in the study, who were subsequently assigned randomly to two groups using a randomization program that also assigned them numbers. Each group was intended to consist of 20 participants. The random allocation software 2.0 was used. 

### 2.3. Procedures and Interventions

All those included in the study kept a 4-week headache diary ahead of the trial, and these diaries were used to determine the number of headache days, the headache durations, and numeric rating scale (NRS) pain scores. After 4 weeks of screening, the patients were divided randomly into the GON block group (*n* = 21) and the transnasal SPG block group (*n* = 20) and continued their prophylactic migraine treatment. All interventional procedures were performed by the same pain medicine and neurology specialist (HAÜ). Patients received GON and transnasal SPG blocks once a week for the first month, and once a month in the subsequent 2 months. All patients were observed for 30 min following the application of the block for possible complications. 

GON Block: The patients were placed in the prone position following intravenous access and monitoring. The one-third medial point along the line between the external occipital protuberance and the mastoid process was palpated. Following antisepsis, 2 cc of 2% lidocaine was injected into the palpated area, via negative aspiration.

Transnasal SPG Block: The patients were placed in the supine position following intravenous access and monitoring. A 2 cc 2% lidocaine-soaked cotton swab was advanced from the nostril along the superior margin of the middle turbinate until it reached the posterior wall of the nasopharynx. When we perceived the sensation of touch, we ceased advancing the cotton swab. The cotton swab was kept in the target area for 20 min. 

### 2.4. Patient Evaluation and Outcomes

A neurologist blinded to the treatment was assigned to evaluate the patients at all visits, (AB) whose age, sex, comorbid diseases, education status, marital status, duration of migraine, prophylactic medications (beta-blockers, antidepressants, flunarizine, angiotensin-converting enzyme (ACE) inhibitors, others), acute symptomatic medication use (paracetamol, nonsteroidal anti-inflammatory drugs (NSAIDs), triptans) were recorded. The quantity of acute medication used (paracetamol, NSAIDs, triptans) was recorded during the first, second, and third-month visits based on headache diary data. The average values of the NRS scores and durations of migraine attacks documented in the headache diaries were calculated.

The patients were subjected to the migraine disability assessment (MIDAS) questionnaire before treatment and at the third-month visit to measure the level of disability associated with headache and to determine the functional outcomes of migraine. The five questions posed by MIDAS acquire data on missed school/work days, household chores, and family, social, and leisure activities. The questionnaire garners data on missed activities and days on which productivity was reduced by at least half, with higher MIDAS scores indicating a decrease in functionality.

### 2.5. Statistical Analysis

Data were analyzed using IBM SPSS Statistics (Version 29.0. IBM Corp.: Armonk, NY, USA), with descriptive statistics presented as numbers (*n*), percentages (%), and mean ± standard deviation, median, minimum, and maximum values. The normality of the distribution of the numerical variables was assessed using a Shapiro–Wilk normality test, and the homogeneity of any variances between the groups was analyzed with a Levene test. For numerical variables, the data from the GON and SPG groups were compared with an independent samples *t*-test if normally distributed, and a Mann–Whitney U test if not. Within-group comparisons of pre-treatment values with those recorded in the first, second, and third months following treatment commencement were conducted using a Friedman analysis, with the Bonferroni correction applied for multiple comparisons. A Wilcoxon test was used for the comparison of the baseline and final MIDAS scores. Chi-square analyses (Yates Chi-square, Fisher–Freeman–Halton exact test) were utilized for the between-group comparison of categorical variables. A *p*-value < 0.05 was considered statistically significant.

## 3. Results

A total of 37 patients were evaluated for the study, with 19 (51.4%) in the GON block group and 18 (48.6%) in the SPG block group (Figure 1). Among the participants, 24 (64.9%) were female and 13 (35.1%) were male, aged 19–52 years, with a mean age of 34.4 ± 8.0 years. Of the total, 16 patients had comorbid diseases, 26 (70.3%) had a university degree, and 19 (51.4%) were married. All patients were receiving prophylactic medical treatment for at least 6 months. The medical prophylactic treatments that had previously failed were not recorded. Table 1 shows the baseline demographic and clinical characteristics of the groups, which did not differ to any statistically significant degree (*p* > 0.05).

Table 2 details the changes recorded in the headache diaries and the MIDAS scores over the course of the treatment. While there was no significant difference in headache intensity between the two groups at baseline, headache intensity was noted to decrease significantly in the GON block at all follow-up visits compared to the baseline, while in the SPG block group, headache intensity decreased significantly in the second and third months. The headache duration and number of headache days in the pretreatment period were much higher in the GON block group. In both groups, a significant reduction in headache duration and the number of headache days was noted during all follow-up visits when compared to the baseline. The MIDAS scores did not differ statistically between the two groups at baseline, and there was a significant decrease in the MIDAS scores of both groups in the third month.

The GON block group experienced a greater reduction in headache intensity, duration, number of headache days, and MIDAS scores than the SPG block group in the third month (Table 2). The number of days on which NSAIDs were used was statistically higher in the GON block group during the pretreatment period, and while there was a significant reduction in the number of days on which NSAIDs were used compared to the baseline values in both groups, this difference was greater in the GON block group (Table 3).

In the SPG block group, two patients developed nasal irritation, and two developed a temporary difficulty in swallowing, while in the GON block group, one patient developed a vasovagal reaction. No major adverse effects were noted. The patients are still being followed up at our outpatient clinic.

## 4. Discussion

The present study found that repetitive GON and SPG blocks decreased both the intensity and duration of headaches, the number of headache days, and the need for acute attack treatments, while also improving functionality. Patients treated with GON blocks experienced statistically greater reductions in headache intensity, durations, number of headache days, MIDAS scores, and the number of acute attack treatments compared with those treated with SPG blocks. Although the number of headache days and acute attack treatments was higher in the GON block group at baseline, they were significantly lower at the 3-month follow-up. The average headache duration was initially longer in the GON block group but was noted to be statistically lower in the SPG block group at the first-month follow-up, while there was no significant difference between the two methods at the second- and third-month follow-ups.

The efficacy of SPG and GON blocks for migraine treatment has been investigated in various studies, yet only a small proportion of these studies were placebo-controlled. Most studies of SPG blocks have focused on their effect on acute migraine attacks, while fewer have assessed their efficacy in prophylactic treatment. Cady R.K. et al. compared SPG block using local anesthetics twice a week for 6 weeks with a placebo for the treatment of chronic migraine, and reported insignificant reductions in headache days, headache intensity, and acute medication use but attributed their findings to the small sample size (*n* = 38). The study compared values in the first and sixth post-treatment months [12]. In contrast, the present study revealed significant reductions in the NRS values in the second and third post-treatment months in those treated with SPG blocks. In another study, the same researchers applied SPG block 12 times over 6 weeks for the treatment of acute attacks in patients with chronic migraine, examined its efficacy in a placebo-controlled study, and reported a decrease in pain intensity within the first 24 h and a decrease in pain intensity compared to the baseline value before each SPG block. In the 4-week post-treatment period, no difference was noted in the use of medication for the treatment of acute attacks when compared to the placebo group [13]. In our study, the patients who were treated with SPG block used fewer NSAIDs to treat acute attacks during the 3-month follow-up compared to the baseline values. In a recent study, the headaches of patients administered SPG block twice a week for 6 weeks for the treatment of chronic migraine and medication overuse were evaluated, revealing a decrease in the number of days with moderate/severe headaches per month and an increase in functionality [11].

There have to date been more studies of GON block than SPG block. In randomized controlled trials, GON block has been found to reduce headache duration and intensity, as well as the number of headache days [16,18]. Chowdhury D et al. reported that GON block with 2% lidocaine once a week for 4 weeks was superior to placebo in reducing the average number of headache days for 3 months in patients with chronic migraine [16]. Inan et al. reported that migraine intensity and duration were reduced and the number of migraine days improved with GON block administered once a week for 4 weeks, and then monthly for two months [18]. Ulusoy E.K. et al. reported an increase in functionality at the end of a 3-month course of GON block treatment, concurring with the findings of the present study [17]. In another study in which GON block was applied once a week for 4 weeks, headache intensity, the number of attacks, and the level of disability were lower even after 6 months compared to the pretreatment values [19]. In a recent meta-analysis, it was reported that GON block reduced headache intensity and the number of analgesics used, but had no effect on headache duration [20]. Although a reduction in headache duration was noted in our study, the results may be misleading as the study was not placebo-controlled.

The SPG is the ganglion with the largest neuronal structure in the calvarium outside the brain [28]. It is located at the posterior attachment of the middle turbinate above the ethmoidal crest [13]. Postsynaptic parasympathetic fibers from the SPG, which is also associated with the maxillary branch of the trigeminal nerve, activate the superior salivatory nucleus. SPG activation triggers the release of acetylcholine, nitric oxide, and vasoactive intestinal peptide, resulting in cerebral vasodilation and neurogenic inflammation. Migraine is known to be triggered by odor stimuli, food, sleep deprivation, and stress. Serial SPG block is thought to inhibit the activation of the superior salivatory nucleus that is activated by such triggers and prevents the superior salivatory nucleus from stimulating the SPG [11,26,27,29,30]. The SPG can be accessed using a subzygomal, intranasal, or intraoral approach, and while the intraoral approach is generally preferred by dentists, in the subzygomal approach, local anesthetic is administered through the use of a cannula under fluoroscopic guidance, being a more invasive technique that requires experience and has a relatively high risk of complications. In the transnasal approach, the SPG can be accessed using a cotton swab or devices such as Spenocath, and is generally a cheaper and easier approach to the administration of local anesthetic to the 1–2 mm thick mucosa associated with the SPG [13,21,31]. A previous study reported that sympathetic block occurred due to the passage of local anesthetic into the pharynx in the drip method using such devices, whereas parasympathetic inhibition developed more in the topical method using a cotton swab [32]. This is the reason we chose a topical application for the present study.

Trigeminal and cervical nociceptive afferents synapse in the trigeminal nucleus caudalis in the brainstem. GON block aims to inhibit the trigeminocervicovascular system and prevent the release of neuroinflammatory mediators, neurogenic inflammation, and cerebral vasodilation activation [33,34]. GON block can be performed proximally at the C2 level under ultrasound guidance, or distally along the nerve’s scalp trajectory using anatomical landmarks, and the two approaches have been shown to have similar effects. Karaoğlan M. et al. reported the proximal approach to be more effective on the number of days with headache [35,36]. In the present study, we opted for the distal technique due to its feasibility in an outpatient setting without the need for ultrasound guidance.

Some demographic factors are reported to be influential in the development of certain headache syndromes. Research indicated that marital status, female gender, comorbidity, and low educational level contribute to the chronicity of migraine [37,38]. In some studies, however, data suggest that there is no association between migraine and socioeconomic factors [39,40]. In our study, marital status, education, and gender did not exhibit differences in influencing treatment outcomes. The full benefit of agents used in migraine prophylactic treatment may take 2–6 months to manifest [41]. In our study, the fact that all patients had been unresponsive to previous medical treatments and had been using their current medical treatments for at least 6 months suggests that the change in headache frequency in their daily headache diaries was likely attributable to the block treatments.

The major limitation of the study is the absence of a placebo-controlled design and the small sample size. Migraine is known to exhibit fluctuations in pain severity and duration, and the placebo effect is considerable in terms of pain management. The patients in the GON block group in the present study had longer headache durations, more headache days, and greater use of medical treatments for acute attacks at baseline. Although all patients were receiving prophylactic medical treatment, no assessment was made of the number of prophylactic treatment failures, suggesting potential differences in treatment resistance between the GON and SPG block groups, which could be considered a limitation of the study.

## 5. Conclusions

Repetitive GON and SPG blockade reduces headache duration, intensity, the number of headache days, and the need for acute medical treatment in patients with episodic migraine. Both treatments are easy to apply, effective, and safe. This randomized study suggests that repetitive GON block provides a superior effect to SPG block, as determined by a significant reduction in migraine days, severity, and duration. However, the sample size is very small and multicenter studies with larger sample sizes will be needed for further evaluation.

## Figures and Tables

**Figure 1 jcm-13-03027-f001:**
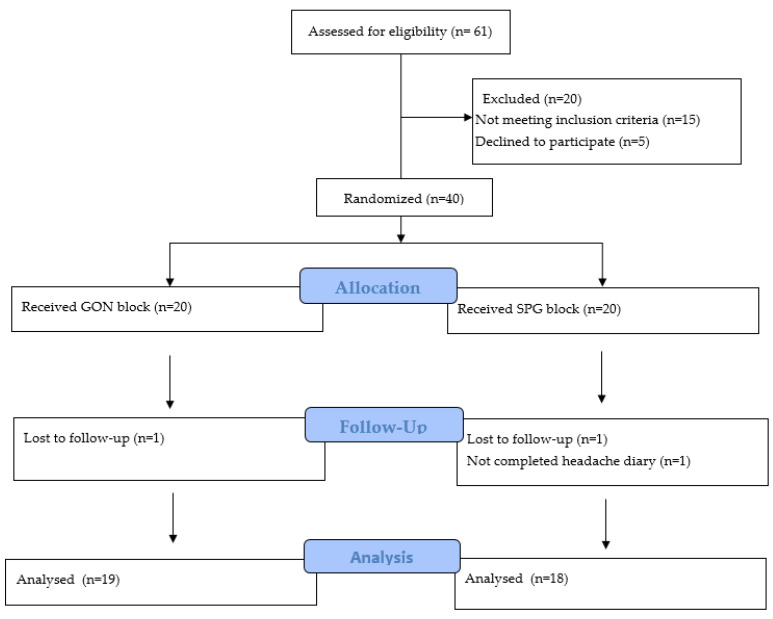
Flowchart of the participating patients.

**Table 1 jcm-13-03027-t001:** Demographic and clinical characteristics of the patients. Abbreviations: GON, greater occipital nerve; SPG, sphenopalatine ganglion; med, medicine; TCA, tricyclic antidepressants; SSNRI, selective serotonergic noradrenergic reuptake inhibitors; VPA, valproic acid; TPM, topiramate; *n*: patient number; %, percentage; mean age ± standard deviation).

Variables	Groups				Test Statistics	
	GON Block *n* = 19		SPG Block *n* = 18		Test Value	*p*-Value
Age	33.5 ± 7.6		35.3 ± 8.5		0.680	0.501 ^†^
Gender, *n* (%)						
Female	13	68.4	11	61.1	0.015	0.904 ^Φ^
Male	6	31.6	7	38.9		
Education *n* (%)					3.478	0.150 ^¥^
University	11	57.9	15	83.3		
High school	7	36.8	2	11.1		
Primary school	1	5.3	1	5.6		
Marital status, *n* (%)					0.948	0.746 ^¥^
Single	7	36.8	9	50.0		
Married	11	57.9	8	44.4		
Divorced	1	5.3	1	5.6		
Prophylactic med, *n* (%)					3.975	0.614 ^¥^
Beta-blockers	5	26.3	6	33.3		
Ca channel Blockers	4	21.1	7	38.9		
TSA	3	15.8	3	16.7		
SSNRI	4	21.1	1	5.6		
VPA	2	10.5	1	5.6		
TPM	1	5.3	0	0.0		

^†^: Independent samples *t*-test in independent variables, ^Φ^: Yates Chi-square test, ^¥^: Fisher–Freeman–Halton Chi-square test.

**Table 2 jcm-13-03027-t002:** Comparison of headache diary and MIDAS scores at baseline and follow-up visits. Abbreviations: GON, greater occipital nerve; SPG, sphenopalatine ganglion; MIDAS; migraine disability assessment. Data are expressed as median (min–max).

		Groups		Test Statistics	
		GON Block	SPG Block	Test Value	*p* Value ^&^
Intensity	Baseline	7.0 (5.0–8.0)	6.0 (4.0–8.0)	1.286	0.198
	1st month	4.0 (2.0–7.0) *	5.5 (4.0–6.0)	3.525	**<0.001**
	2nd month	4.0 (3.0–5.0) *	5.0 (4.0–6.0) *	3.847	**<0.001**
	3rd month	4.0 (2.0–4.0) *	5.0 (4.0–6.0) *	5.041	**<0.001**
	Difference (baseline-3rd month)	3.0 (1.0–4.0)	0.0 (−1.0–3.0)	4.608	**<0.001**
	Test value; *p* value ^‡^	43.082; **<0.001**	15.270; **0.002**		
Duration	Baseline	24.0 (6.0–48.0)	18.0 (4.0–48.0)	1.972	**0.049**
	1st month	6.0 (3.0–24.0) *	4.0 (2.0–24.0) *	2.397	**0.017**
	2nd month	4.0 (1.0–8.0) *	4.0 (3.0–12.0) *	1.279	0.201
	3rd month	3.0 (1.0–6.0) *	4.0 (2.0–12.0) *	1.949	0.051
	Difference (baseline-3rd month)	20.0 (2.0–47.0)	14.0 (0.0–36.0)	2.297	**0.022**
	Test value; *p* value ^‡^	47.873; **<0.001**	43.575; **<0.001**		
Day	Baseline	6.0 (4.0–10.0)	4.0 (3.0–8.0)	2.398	**0.016**
	1st month	1.0 (0.0–3.0) *	3.0 (1.0–4.0) *	4.267	**<0.001**
	2nd month	1.0 (0.0–3.0) *	2.0 (1.0–4.0) *	4.362	**<0.001**
	3rd month	1.0 (0.0–3.0) *	3.0 (1.0–4.0) *	4.691	**<0.001**
	Difference (baseline-3rd month)	5.0 (3.0–9.0)	2.0 (1.0–4.0)	5.018	**<0.001**
	Test value; *p* value ^‡^	47.870; **<0.001**	46.467; **<0.001**		
MIDAS	Baseline	16.0 (8.0–42.0)	15.0 (8.0–24.0)	1.058	0.290
	3rd month	2.0 (0.0–8.0)	9.0 (4.0–14.0)	4.681	**<0.001**
	Difference (baseline-3rd month)	13.0 (4.0–41.0)	6.0 (4.0–12.0)	4.369	**<0.001**
	Test value; *p* value ^†^	3.828; **<0.001**	3.758; **<0.001**		

^&^: Mann–Whitney U test, ^‡^: Friedman test, ^†^: Wilcoxon test, *: indicates measurement intervals that differ from baseline values.

**Table 3 jcm-13-03027-t003:** Comparison of the number of acute medical treatments used at baseline and follow-up visits. Abbreviations: GON, greater occipital nerve; SPG, sphenopalatine ganglion; NSAID: nonsteroidal anti-inflammatory drug. Data are expressed as median (min–max).

		Groups		Test Statistics	
		GON Block	SPG Block	Test Value	*p* Value ^&^
Number of NSAID days	Baseline	6.0 (3.0–8.0)	4.0 (2.0–8.0)	2.482	**0.013**
	1st month	2.0 (1.0–4.0) *	3.0 (1.0–5.0) *	2.189	**0.029**
	2nd month	2.0 (1.0–4.0) *	2.0 (1.0–4.0) *	0.117	0.907
	3rd month	2.0 (1.0–3.0) *	3.0 (1.0–5.0) *	2.769	**0.006**
	Difference (baseline-3rd month)	4.0 (1.0–7.0)	2.0 (0.0–3.0)	4.29	**<0.001**
	Test value; *p* value ^‡^	49.336; **<0.001**	46.050; **<0.001**		

^&^: Mann–Whitney U test, ^‡^: Friedman test, *: indicates measurement intervals that differ from baseline values.

## Data Availability

The data that support the findings of this study are available upon reasonable request from the corresponding author, K.S.C.T. The data are not publicly available due to their containing information that could compromise the privacy of the study participants.

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
