# Peer review of "Comparison of Greater Occipital Nerve Blockade and Sphenopalatine Ganglion Blockade in Patients with Episodic Migraine"

_jcm, 2024, doi:10.3390/jcm13113027_

Round 1

Reviewer 1 Report

Comments and Suggestions for Authors

Dear Authors,

The article is interesting and compares two injectable treatment options for migraine management.

There are some revisions that I kindly ask the authors to review:

1. Explain what is meant by interventional treatment for headache line 87

2. Explain why the included patients are on preventive therapy and why preventive therapy was not washed out.

How long after starting preventative therapy were patients included in the study? Specify to exclude that the outcome of the injection therapy depends on it and not on the oral therapy

3. Why were education and marital status considered among the variables? How do they affect?

4. In the discussion, comparing the current results with other studies, it would be appropriate to explain whether in the other studies the patients treated with GSP were also subjected to oral preventive therapy. At the current state of writing the article, the results of the present study would seem to give a better therapeutic response, without underlining that the patients enrolled were all subjected to oral preventive therapy and that the benefit could be attributable to this rather than to the nerve block.

Reviewer 2 Report

Comments and Suggestions for Authors

Dear Sir, 

          I have the following comments regarding your article

1. Abstract:

o Major limitation is no placebo comparison group invasive vs non-invasive; Among invasive groups drug vs placebo

o    Migraine is known to have fluctuations in pain severity and duration & placebo effect is high in terms of pain.

o    1-2 lines about the methodology of GON/SPG could be given

2. Material and methods

o    Definition of episodic migraine in conventional terminology missing

o    sample size is small

o    Is there selection bias?

o    Are they referral patients or consecutive?

o    Invasive procedures require specialists and frequent in-hospital visits.

3. Results

o    No. of prophylactic drugs given in each group & treatment failure

o    Baseline characteristics of the two groups (GON & SPG) are not matched in terms of severity and duration of pain

o    In the SPG group only 1 point reduction in pain severity, does it mean a substantial reduction (Table)

o    How many patients have total recovery being invasive procedures

o    Follow up after 3 months period

4. Conclusion

o    The concluding remark is too assertive “GON should be” for a sample size of 39 and with baseline disease characteristics in between-group differences.

Reviewer 3 Report

Comments and Suggestions for Authors

It is not easy to present a negative critic to this paper. It is well done.

Introduction: Definition of migraine and prophylactics are presented.

Row 50-57 positive you describe how GON block inhibits the trigeminocervical complex and prevents neurogenic inflammation!

Row 58-64 you give the corresponding background of SPG block.

Material and methods are clearly presented. In SPG blockade was the application guided visually or just ‘by feeling’? Relevant presentation of statistic methods.

I find the presentation in the tables clear.

In discussion you argue for your choices of how to applicate your blockades.

Lack of a specific section about weaknesses and strengths. Maybe it is sufficiently treated in the running text?

The references seem relevant.

Round 2

Reviewer 2 Report

Comments and Suggestions for Authors

Dear Sir,

         I have the following comments regarding your article.

All the suggestions incorporated by the authors, No further comments.